# OMNINET: A UNIFIED ARCHITECTURE FOR MULTI-MODAL MULTI-TASK LEARNING

## ABSTRACT

Transformer is a popularly used neural network architecture, especially for language understanding. We introduce an extended and unified architecture that can be used for tasks involving a variety of modalities like image, text, videos, etc. We propose a spatio-temporal cache mechanism that enables learning spatial dimension of the input in addition to the hidden states corresponding to the temporal input sequence. The proposed architecture further enables a single model to support tasks with multiple input modalities as well as asynchronous multi-task learning, thus we refer to it as *OmniNet*. For example, a single instance of *OmniNet* can concurrently learn to perform the tasks of part-of-speech tagging, image captioning, visual question answering and video activity recognition. We demonstrate that training these four tasks together results in about three times compressed model while retaining the performance in comparison to training them individually. We also show that using this neural network pre-trained on some modalities assists in learning unseen tasks such as video captioning and video question answering. This illustrates the generalization capacity of the self-attention mechanism on the spatio-temporal cache present in *OmniNet*.

## 1 INTRODUCTION

Transformer (Vaswani et al., 2017) is currently one of the best performing models for any sequence transduction tasks, especially those involving natural language. It is originally designed for a single task at a time. In fact, most of the generic deep learning architectures (Wu et al., 2016; Szegedy et al., 2016; Battenberg et al., 2017) that have been designed and developed are able to learn, albeit very well, a single task and handle one task specific input domain like image, text or audio. Furthermore with these models, we often rely on the generalization capability of the trained network to guarantee performance on unseen examples. Transfer learning (Shu et al., 2018; Hu et al., 2018) is another popular paradigm used to adapt the model to learn a related task with similar input domain. The success of neural networks across these challenges is known to be due to their ability in learning effective representations of the data. For example, the self-attention mechanism in Transformers can capture the global temporal dependence in sequential data very well. Naturally, the question arises whether we can extend these architectures, like the Transformer, to be able to learn shared representations from multiple input domains and to be able attend on these representations to perform a multitude of tasks concurrently.

The research into multi-task models that learn to solve varied tasks across a multitude of input domains is not new. Work done in Ngiam et al. (2011) demonstrates an architecture capable of learning a shared representation across audio and video modalities. Similarly in Collobert & Weston (2008) a convolutional architecture has been designed to support a variety of NLP tasks. However, most of these architectures are designed to learn specific set of tasks with known input domains. To the best of our knowledge, there does not exist a single unified architecture that works out of the box for any combination of multi-modal inputs.

To address this gap, we extend Transformer towards a unified architecture, namely *OmniNet*, which enables a single model to support tasks with multiple input modalities and asynchronous multi-task learning. We consider that most real-life data like image, text, speech, video, etc. is a direct conjunction of spatial and temporal components. Therefore, we employ a spatio-temporal cache mechanism to learn a shared representation of the input data across the spatial (space) and temporal (time) di-

mension. Using a generalized *encode()* function, *OmniNet* can process and store spatio-temporal representation for each of the input domains and then *decode()* predictions across a multitude of tasks. In our experiments, we train a single instance of the *OmniNet* to solve a number of tasks spanning multiple multi-domains such as part-of-speech tagging, image captioning, visual question answering and video activity recognition. To make our work reproducible, open to scrutiny and further development, we will open source a demonstration of our system implemented using Pytorch (Paszke et al., 2017).

## 2   RELATED WORK

Multi-task learning has been extensively studied in the literature, with applications to a wide set of problems ranging from natural language processing (NLP) (Collobert & Weston, 2008; Johnson et al., 2017; Dong et al., 2015) to speech recognition (Seltzer & Droppo, 2013; Krishna et al., 2018) to vision (Zhang et al., 2014; Chen et al., 2018; Anderson et al., 2018). It has also found its use in a combination of diverse tasks like image captioning and text translation and parsing (Luong et al., 2016; Zhao et al., 2018). However, most of these architectures assume the set of tasks to be known in advance. Similarly, multi-modal learning has been essential for solving a broad range of interesting problems such as Visual Question Answering (Kim et al., 2016; 2018) and Video Question Answering (Lei et al., 2018). Again, the state-of-the-art models are highly specific to the objective in hand and not easily adaptable to different tasks or domains. (Kaiser et al., 2017) proposed MultiModel architecture for learning multiple tasks but lacks support for multi-modal tasks with more than one input domains such as visual question answering.

## 3   PROPOSED MODEL

We propose a unified architecture, namely *OmniNet*, to enable learning multi-modal tasks with multiple input domains and support generic multi-tasking for any set of tasks. The *OmniNet* architecture consists of multiple sub-networks, called peripheral networks, connected to a common central neural network called the Central Neural Processor (CNP) (Figure 1). Each peripheral network is used to encode the domain specific input into feature representations. In this work, we describe image, text and video peripherals (Section 3.1). One can add more, say speech peripheral, depending on the task. The output representation of a peripheral network is always a spatio-temporal tensor $x \in \mathbb{R}^{t \times s \times d_{model}}$, where $t$ & $s$ are the temporal and spatial dimensions of the input respectively, and $d_{model}$ is the model dimension input to the CNP.

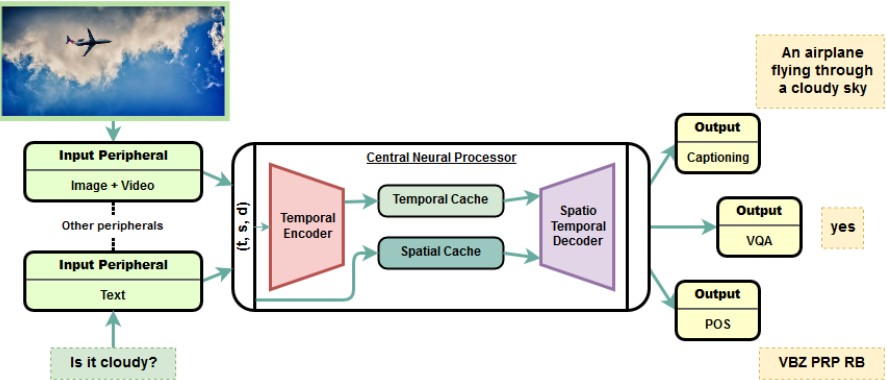

Figure 1: *OmniNet* performing image captioning, visual question answering and POS tagging at once

The spatio-temporal representations generated by the peripheral networks corresponding to each input domain are then processed by the CNP. The CNP uses fully attention based encoder-decoder (Bahdanau et al., 2015; Sutskever et al., 2014; Cho et al., 2014) model for sequence transduction similar to the Transformer architecture (Vaswani et al., 2017), which is the state-of-the-art for multiple language modeling tasks (Section 3.2). During the encoding stage, the CNP implements a

generic *encode(x, D)* function to first process and store the spatio-temporal representations of the input, where $x \in \mathbb{R}^{t \times s \times d_{model}}$ is the spatio-temporal tensor produced by the peripheral networks and $D \in \mathbb{Z} : 0 \leq D < D_{len}$ is the domain id and $D_{len}$ is the max number of domains supported by the CNP. The *encode()* function is called multiple times, once for each multi-modal input from respective peripheral. During the decoding stage, a *decode($y_{shifted}$, $\tau$)* function is used to decode predictions as softmax probabilities, where $y_{shifted} \in \mathbb{Z}^{N-1}$ are the target outputs shifted one time-step to the right, $N$ is the length of the output sequence; $\tau \in \mathbb{Z} : 0 \leq \tau < \tau_{len}$ is task id and $\tau_{len}$ is the total number of supported tasks. The decoding step is similar to Vaswani et al. (2017), modified to incorporate a two-step attention mechanism over spatial and temporal cache.

## 3.1 PERIPHERAL NETWORKS

First, we elaborate on how we support multiple input domains using peripheral networks. A peripheral network can use a pre-trained model from existing literature to ultimately encode a given domain input to a standardized feature representation $x \in \mathbb{R}^{t \times s \times d_{model}}$, where $t$ & $s$ are the temporal and spatial dimensions of the input respectively, and $d_{model}$ is the model dimension input to the Central Neural Processor. Here we detail text and vision peripherals and one can add more peripherals or alter the peripheral design depending on the task.

**Vision peripheral:** This peripheral uses a convolutional neural network to encode image and video inputs in the tasks. For an image of dimension $h \times w \times n_c$, this peripheral down-samples in to $h' \times w' \times n_c'$, where $h, w, n_c$ are the height, width and number of input channels respectively. For a video, each frame is input to the peripheral to produce $F \times h' \times w' \times n_c'$, where $F$ is the total number of frames in the video. The encoding vectors are then projected to dimension $d_{model}$ using a fully connected layer. The output is then reshaped into a spatio-temporal tensor of $x \in \mathbb{R}^{t \times h'w' \times d_{model}}$, where $t = 1$ for an image and $t = F$ for a video. In our experiments, we use the pre-trained *ResNet-152* model, a variant of ResNet (He et al., 2016) consisting of 152 convolutional layers. We remove the final fully connected and avg-pooling layers to generate spatial feature representations for a given image/video.

**Language peripheral:** The Language peripheral uses byte-pair encoding (Sennrich et al., 2016) to generate subwords for a given input sentence. The subwords are passed to an embedding layer to generate subword embeddings of dimension $d_{emb}$ and projected to dimension $d_{model}$ using a fully connected layer. The output is then reshaped into a spatio-temporal tensor $x \in \mathbb{R}^{t \times 1 \times d_{model}}$, where $t$ equal to number of subwords in the input sentence. As we do not have any spatial dimension in textual data, the spatial dimension of $x$ from a Language peripheral is always 1. In our experiments, We used pre-trained subword embeddings with $d_{emb} = 300$ and $vocab\_size = 25000$ from Heinzerling & Strube (2018), which includes pre-trained subword embeddings of over 275 languages, to initialize the weights of the embedding matrix.

## 3.2 CENTRAL NEURAL PROCESSOR (CNP)

To process the spatio-temporal information in the input data, the CNP implements a spatial cache $C_s$, temporal cache $C_t$ and a link array $L$. The spatial and temporal cache and the link array are a list of elements, initialized as empty before the encoding process. During the encoding stage, an *encode()* routine takes as input, the tensor $x$ generated from the peripheral and corresponding domain/peripheral id $D$. This function processes the spatial and temporal information in the input $x$ and stores them into the spatial cache $C_s$ and the temporal cache $C_t$, respectively and stores their dimensions $t$ & $s$ in the link array. For a given task, this *encode()* routine is called $K$ times, where $K$ is the number of inputs in the task. Note that these inputs can belong to same or different domains.

**Encode ($x$, $D$):** For a given input $x \in \mathbb{R}^{t \times s \times d_{model}}$ and domain identifier $D$, the *encode()* routine is described in Algorithm 1. Since inputs can come from multiple peripherals, the algorithm first concatenates the input with the domain embedding to ensure a domain-aware encoding of the input (Steps 2 to 3). Steps 4 to 7 process the spatial information in $x$ by unrolling the time dimension and adding these unrolled vectors into the spatial cache. Steps 8 to 10 process the temporal information in $x$ by averaging the spatial dimension of $x$ and then passing the averaged tensor to a self-attention based *TemporalEncoder*. This *TemporalEncoder* is similar to the encoder used in Vaswani et al. (2017) as shown in Figure 2 is used to calculate temporal embeddings of the input sequence. The output from the *TemporalEncoder* is appended to the temporal cache.

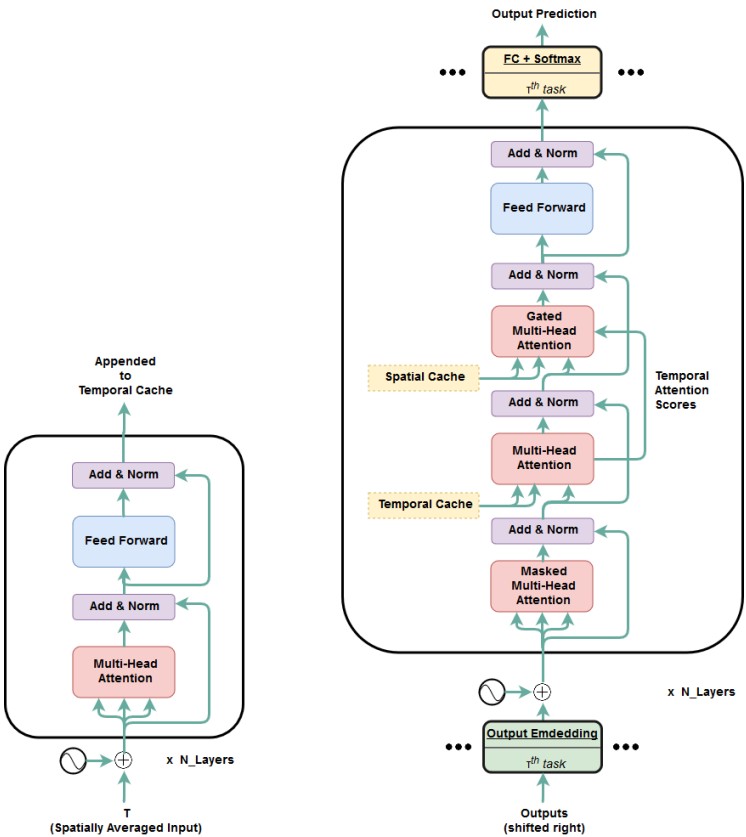

Figure 2: **left:** *TemporalEncoder* architecture; **right:** *OmniNet decode()* architecture.

---

**Algorithm 1** *encode()*: Encodes spatial and temporal representations into spatial and temporal cache

---

**Require:** $x \in \mathbb{R}^{t \times s \times d_{model}}, D, C_s, L, C_t$
1: $L \leftarrow L \cup (t \rightarrow s)$
2: $D_{emb} \leftarrow EmbedLayer(D)$
3: $x \leftarrow FC(Concat(x, D_{emb}), d_{model})$
4: **if** $s > 1$ **then**
5:      $S \leftarrow Reshape(x, (ts, d_{model}))$ {where, output $S = [S_1, \ldots, S_{ts}]$ s.t. $S_i \in \mathbb{R}^{d_{model}}$ is a spatial feature vector.}
6:      $C_s \leftarrow C_s \cup [S_1, \ldots, S_{ts}]$ {Append spatial representations to spatial cache}
7: **end if**
8: $T \leftarrow (\sum_{i=1}^{s} x[:, i, :])/s$
9: $T \leftarrow TemporalEncoder(T)$ {where, output $T = [T_1, \ldots, T_t]$ s.t. $T_j \in \mathbb{R}^{d_{model}}$ is the encoding of temporal dimension in $x$.}
10: $C_t \leftarrow C_t \cup [T_1, \ldots, T_t]$ {Append temporal representations to temporal cache}

---

The above encoding routine keeps appending spatio-temporal information to $C_t$ & $C_s$ for each input $x_k \in \mathbb{R}^{t_k \times s_k \times d_{model}}$. Note the superscript $k$ to denote correspondence to $k$-th input of the task, where $k \in 1, \ldots, K$. After $K$ calls, we have the temporal cache $C_t = [T_1, \ldots, T_R]$, where $R = \sum_{r=1}^{K} t_r$; the spatial cache $C_s = [S_1, \ldots, S_P]$, where $P = \{\sum_p t_p * s_p : p \in 1, \ldots, K \wedge s_p > 1\}$ and the link array $L = [(t_1 \rightarrow s_1), \ldots, (t_K \rightarrow s_K)]$. Note that $C_s$ can also be empty in case the *encode()* is only called with inputs with $s_k = 1 \forall k$. Next, we use the *decode()* routine to generate predictions as softmax probabilities.

**Decode** $(y_{shifted}, \tau)$: The architecture of the *decode()* function is shown in Figure 2. The *decode()* takes as argument the output labels $y_{shifted}$ shifted one time step to the right, a task id $\tau$ and gen-

erates predictions by attending from the spatial and temporal cache. The *decode()* function is structured similar to the decoder used in the *Transformer* architecture Vaswani et al. (2017) and jointly attends on the vectors stored in the temporal and spatial cache. Similar to Vaswani et al. (2017), the decoding first starts by attending over the output embeddings using masked multi-head scaled dot product attention. The attention layer for the temporal cache uses scaled dot-product attention with multiple heads as specified in Vaswani et al. (2017). Attention layer for the spatial cache, uses gated multi-head attention to attend over the elements of the spatial cache. For inputs with both time and space dimension (e.g. video), we want the spatial attention layer to attend more on frames which have relatively high attention scores in the temporal cache attention layer. Therefore, the attention score output from the temporal cache multi-head attention layer $A \in \mathbb{R}^{n_h \times N \times R}$, is used to calculate the tensor $G \in \mathbb{R}^{n_h \times N \times P}$ used for gating the attention score output in the spatial attention layer, where $n_h$ is the number of heads in multi-head attention as described in Vaswani et al. (2017). The tensor $G$ is calculated using $A$ & $L$ as detailed in Algorithm 2. Given[1] $Q$: the matrix of queries, $K$: keys of dimension $d_k$ and $V$: values of dimension $d_v$, the scaled dot-product attention for the spatial layer is modified as:

$$\text{Attention}(Q, K, V, G) = \left( \text{Softmax}\left(\frac{QK^T}{\sqrt{d_v}}\right) \odot G \right) V \tag{1}$$

In order to use the same CNP for multiple tasks with varying output vocabularies, we use multiple output embedding layers $OutputEmbedLayer_1, \ldots, OutputEmbedLayer_{\tau_{len}}$, to generate the output embeddings for each task. At the final layer, we use multiple $(FC + Softmax)_1, \ldots, (FC + Softmax)_{\tau_{len}}$ classification layers for each task. We also calculate a task embedding vector using $\tau$ and always start decoding using the task embedding vector.

---

**Algorithm 2** Calculate $G$ using output scores from temporal attention and link array

**Require:** $L, A$
1:   $idx \leftarrow 0$
2:  **for each** t, s in L **do**
3:     $G \leftarrow []$
4:     **if** $s > 0$ **then**
5:       $A' \leftarrow A[:, :, idx : idx + t]$
6:       $A' \leftarrow Expand(A', (n_h, N, t, s))$ {where $Expand(tensor, dimension)$ expands tensor according to a given dimension}
7:       $A' \leftarrow Reshape(A', (n_h, N, ts))$
8:       $G \leftarrow G \cup A'$ {Append the respective temporal attention scores to $G$}
9:     **end if**
10:   $idx \leftarrow idx + t$
11: **end for**
12: $G \leftarrow Stack(G)$ {Stack the list of tensors to construct tensor $G$ of dimension $(n_h, N, P)$}

---

### 3.3 MULTI-TASK LEARNING

In order to train a single a model simultaneously on mutiple tasks we used the HogWild training approach as described in Recht et al. (2011). Similar to the approach described in Mnih et al. (2016), the main process holds a global copy of the model. We create separate worker processes for each task, where each process maintains a local copy of a model. At each training iteration, each process starts by synchronizing its local model with the global copy. This is done through forward and backward propagation on its local copy and then copying the locally computed gradients to the global model asynchronously. Each process then calls the global model optimizer asynchronously to update the weights of the global model. Instead of storing the model in CPU as in Mnih et al. (2016) we always store the local copies across multiple GPUs.

## 4 TASKS AND SETUP

To evaluate the effectiveness of our proposed framework for tasks spanning diverse modalities, we choose a set covering all possible spatio-temporal data archetypes: Image Captioning, Part-of-

---

[1]For brevity, we reuse the notations of Vaswani et al. (2017) in this description

Speech (POS) tagging, Visual Question Answering (VQA) and Video-activity Recognition. Each of these tasks explores a unique potential spatio-temporal configuration of the input containing different values of $t$ and $s$, where $t$ and $s$ are the temporal and spatial dimensions of the input respectively. This enables us to perform a comprehensive study of the multi-modal and multi-input capabilities of the system. We further elaborate on the properties of these tasks below.

For training, we always use cross-entropy loss with Adam optimizer (Kingma & Ba, 2015) and schedule the learning rate using Noam scheduler (Shazeer & Stern, 2018) similar to (Vaswani et al., 2017) [2]. In the vision peripheral, we freeze the layers of pretrained ResNet model. Remaining parts of the architecture (peripherals and CNP) are kept trainable. In this section, we provide details on the datasets used and the model setup for each of these tasks.

**Part-of-speech (POS)Tagging:** To illustrate the task with only temporal modality ($t > 1$ & $s = 1$, where $t$ and $s$ are the temporal and spatial dimensions of the input respectively), we consider POS tagging problem. Given an input sequence of words, the model should produce a sequence of POS tags corresponding to each word. We use Penn Tree-bank[3] (Marcus et al., 1994) which contains gold annotations on English WSJ articles by experts. During the encoding stage, each input sentence is processed by the language peripheral to generate a spatio-temporal tensor $x \in \mathbb{R}^{t \times 1 \times d_{model}}$, where $t$ is the sequence length of subwords in the input. The CNP $encode()$ function is then used to encode $x$ into the temporal cache. Note that spatial cache is empty for text inputs as $s = 1$. Therefore, the decoding stage is same as that of Transformers to predict the sequence of POS tags.

**Image Captioning:** This task represents the ones with inputs containing only spatial modality ($t = 1$ & $s > 1$). The captioning model is required to predict text caption for a given image. We use the MSCOCO 2014 dataset (Lin et al., 2014) for training and present results on the COCO validation set. During the encoding stage, the input image is resized to $224 \times 224$ and processed by the vision peripheral containing pre-trained ResNet-152 to produce image embeddings $x \in \mathbb{R}^{1 \times 49 \times d_{model}}$. $x$ is then input to the *encode()* function which populates corresponding spatial and temporal cache. The decoding stage uses *decode()* function with output vocabulary size 25000 to generate the captions.

**Visual Question Answering:** For the task with inputs from multiple domain, such that each contains either spatial or temporal modality (either $t > 1$ & $s = 1$, or $t = 1$ & $s > 1$ for each input), we choose the task of visual question answering. Given a question over an image as inputs, the model is supposed to predict the correct answer label. We use the recently introduced VQA v2.0 dataset (Goyal et al., 2017) for this purpose and perform evaluation on the VQA test-dev set. All the images are resized to dimension $224 \times 224$ before training. The encoding stage of this task utilizes two peripherals: the vision peripheral is used to generate a tensor $x_1 \in \mathbb{R}^{1 \times 49 \times d_{model}}$ for the input image. The language peripheral is used to encode the questions into $x_2 \in \mathbb{R}^{t \times 1 \times d_{model}}$, where $t$ is equal to the length of the subwords in the question. The *encode()* function is the called two times, first with $x_1$ and second with $x_2$ as input. Finally, the $decode()$ with output vocabulary size 3500 is to generate the answers as softmax probabilities in a single decoding step.

**Video Activity Recognition:** For tasks which contain both spatial and temporal modality in a single input ($t > 1$ & $s > 1$), we consider the action recognition task on videos. For this purpose, we use the HMDB dataset (Kuehne et al., 2011). The dataset consists of over 5000 short length clips of real life actions with 51 classes. We present our results on train-test split 1. We use 16 frames per video and resize each of them to $224 \times 224$. During the encoding stage, each frame of the video is passed through the vision peripheral to cumulatively generate a video encoding $x \in \mathbb{R}^{16 \times 49 \times d_{model}}$ which is then used as input to the *encode()* function. Finally, the $decode()$ with output vocabulary size 51 is to predict the action as softmax probabilities in a single decoding step.

## 5 RESULTS AND DISCUSSION

We present the evaluation on (a) Tasks of various modalities illustrated in Section 4 (b) Multi-tasking setup for these tasks (Table 1) (c) Reuse of the multi-task model for an unseen task (Figure 3). In addition, we also provide some ablation studies on the architecture (Table 2)

---

[2]The hyperparameter values used for $n_h$, $d_{model}$, $N\_Layers$, $d_k$, $d_v$ are same as that specified in Transformer base model (Vaswani et al., 2017).

[3]https://catalog.ldc.upenn.edu/LDC99T42; We use splits 0-18 as training, 19-21 as development and 22-24 as test sets

**Performance of proposed architecture on individual tasks:** We choose a set of four tasks with diverse input modalities and combinations as described in previous Section 4. We train the *OmniNet* model independently across each of the above tasks. Each of the tasks demonstrates unique capabilities of this generic architecture. More specifically, in Table 1 we compare our results with the following state-of-the-art[4]:- POS tagging: Spoustová et al. (2009); image captioning & VQA: Anderson et al. (2018) and HMDB: Simonyan & Zisserman (2014). It is important to note that we do not perform any hyper-parameter optimization. We believe that, with more computational power, fine tuning the hyperparameters towards these tasks should result in comparable or even improved performance to the state-of-the-art. These results can indeed be used as a baseline for any future work which aims at using a single architecture across various possible spatio-temporal archetypes. It is interesting to note that the model is extensible to a new domain without any modification to the CNP as long as one can add a specific peripheral to convert domain inputs into spatio-temporal tensors. This aspect of the architecture makes it applicable to several popular multi-modal tasks.

| | POS | Captioning | | Visual Question Answering | | | | HMDB | |
|---|---|---|---|---|---|---|---|---|---|
| | Acc. | BLEU-4 | Meteor | Overall | Y/N | Num. | Other | Acc. | #PARAMS |
| SOTA | 97.44 | 36.2 | 27.0 | 63.2 | 80.3 | 42.8 | 55.8 | 59.4 | - |
| IND | 95.61 | 28.9 | 25.2 | 55.31 | 74.09 | 35.17 | 46.35 | 55.29 | $450m$ |
| MULT-3 | 95.82 | 28.8 | 25.2 | 56.79 | 76.75 | 35.82 | 47.16 | - | $149.03m$ |
| MULT-4 | 95.44 | 27.4 | 24.5 | 55.76 | 75.49 | 35.64 | 46.08 | 54.44 | $149.07m$ |

Table 1: Performance of *OmniNet* on diverse set of tasks. IND: Model trained individually for each of the given tasks; MULT-3: Multi-task model trained on POS, Captioning & VQA; MULT-4: Multi-task model trained across all the four tasks.

**Effect of training a diverse set of tasks together:** We trained two multi-task models: (1) MULT-3 (POS+VQA+Captioning) and (2) MULT-4 (POS+VQA+Captioning+HMDB), using hogwild approach. While the MULT-3 model attains similar and sometimes better performance, the final MULT-4 model attains slightly reduced performance, when compared to the independent task scores. We believe this is due to the skewness in the size of the HMDB dataset containing only 5000 training samples. However as a tradeoff, adding the HMDB task shows interesting zero-shot results demonstrated below. Using a multi-task model also results in three times reduction in the total number of parameters. That is, when a separate model is used for each task, we have a total of over $450 \times 10^6$ parameters. Whereas during multi-tasking since a single model is shared, we have a total of over $149 \times 10^6$ parameters, while achieving similar performance. Interestingly, the model is able to attend on spatio-temporal components of the inputs from different tasks and concurrently generate predictions across them, thus demonstrating the generalization capability of our architecture.

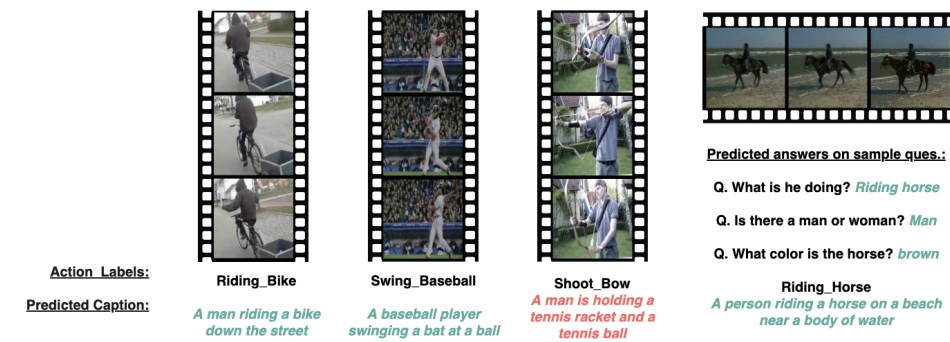

Figure 3: Results of zero-shot video captioning and video question-answering.

**Towards zero-shot learning: reuse of pre-trained network for unseen task:** Sharing representations across multiple tasks provides the benefit to transfer of useful knowledge across multiple domains. Since, image and video are processed by the same vision peripheral, we conducted an experiment to see whether our model pre-trained on all the four tasks (MULT-4) can perform video

---

[4]Since most of these tasks are popular challenges, we compare with state-of-the-art which are generically applicable for the respective task instead of the challenge dataset.

captioning and video question-answering without any explicit training on these tasks i.e. zero-shot learning. The results of the evaluation on randomly picked instances from the HMDB test split 1 are shown in Figure 3. Interestingly, the model performs quite well on related actions that were present in the COCO and VQA training set; such as captions related to horse riding and baseball; or questions related to concepts present in VQA. Without training on any video captioning & video QA instance, the model could use the trained information from image captioning, image QA (VQA) and video action recognition (HMDB) apply them on videos to generate meaningful predictions, hence demonstrating the capability of the model to transfer knowledge across related multi-modal tasks. However, on concepts that are not present in the trained datasets, the model either describes the environment in the video or replaces with alternate known concepts. This case study, although not comprehensive, shows the capability of the model to learn shared representations and ability to transfer knowledge across domains. We believe that adding more tasks and domains will lead to more interesting zero-shot learning results in future across a wide range of problems.

**Impact of individual architectural components:** In order to support different input domains, our architecture introduces spatial cache and link array components to the original *Transformer* architecture (which only consists of mechanisms to handle temporal data). We conducted an ablation study on each of these components to verify their importance across various tasks as shown in Table 2. The ablation was conducted on the independent (IND) as well the multi-tasking model (MULT-4). The second row ablates the link array from our architecture i.e. removing the multiplication of $G$ in Equation 1. The link array was designed to assist in tasks with inputs such as video, containing both spatial as well as temporal modality in a single input. The total number of spatial components becomes very large as number of frames in the video increases, thereby making it difficult to attend on various spatial regions throughout the video. Using link array the spatial attention layer can attend more on specific important frames in the video. Therefore, removal of link array leads to a huge reduction in performance in HMDB compared to other tasks as they do not have both spatio-temporal modalities for any single input. Removal of spatial cache, on the other hand, has significant effect on performance across all tasks containing spatial modality. Since, image captioning contains primarily spatial modality and hence the BLEU drops significantly after ablation. As other tasks utilize the temporal cache for prediction, in the multi-task setting the captioning task learns to utilize the spatial average of the image stored in the temporal cache for prediction and hence retains some performance after ablation of the spatial cache. On the other hand when trained independently on captioning, the network learns to utilize the information in the spatial cache only, and hence the score drops to zero after ablation. VQA leverages both, spatial information from image and temporal information from question, retains some performance from the use of temporal cache. Note that, POS tagging task is not affected by ablation of any of the components since it only has temporal modality in the input.

| | **POS** (Acc.) | | **Captioning** (BLEU-4) | | **VQA** (Overall) | | **HMDB** (Acc.) | |
|---|---|---|---|---|---|---|---|---|
| | IND | MULT-4 | IND | MULT-4 | IND | MULT-4 | IND | MULT-4 |
| *OmniNet* | 95.61 | 95.44 | 28.9 | 27.4 | 55.31 | 55.76 | 55.29 | 54.44 |
| w/o link array | 95.61 | 95.44 | 28.9 | 27.4 | 54.05 | 55.30 | 45.94 | 46.79 |
| w/o spatial cache | 95.61 | 95.44 | 0 | 11.9 | 39.86 | 44.24 | 10.91 | 11.50 |

Table 2: Ablation study on the effect of proposed architectural components.

# 6 CONCLUSIONS AND FUTURE WORK

We present a unified neural network architecture *OmniNet* capable of learning tasks with multiple inputs of varying modalities. The architecture can be further adopted for multi-task learning across any set of tasks containing spatio-temporal data. Sharing one model across multiple tasks also results in a significant reduction in the total number of parameters. We further demonstrate that this shared model can learn robust representations from various spatio-temporal inputs which are reusable for unseen tasks. We believe that this proposed architecture has wide applicability to any task with spatio-temporal inputs. To extend its usability, we would like to introduce new peripherals supporting more domains such as speech. We are also keen on exploring other aspects to the data beyond temporal and spatial dimensions such as graphs and relational data. Further, it would be interesting to investigate scheduling mechanisms for optimizing the multi-tasking framework.

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
