# OpenReview forum: "OmniNet: A unified architecture for multi-modal multi-task learning"
_ICLR.cc/2020/Conference — Reject_

### Official Review · AnonReviewer1 · 2019-10-22
**Official Blind Review #1**

**Rating:** 1

**Review:**

This paper describes the "OmniNet" architecture, which is essentially a transformer to convert any 2-dimensional (time and spatial) input into a sequence of output tokens. The idea is that a single model (the "Central Neural Processor") would learn to perform multiple tasks on multiple inputs at the same time. This seems like a reasonable approach, and would allow a single model to be able to process text, image, as well as video or potentially speech input just the same. Dedicated modality-specific "input peripherals" would "normalize" the data appropriately, and a "start token" seems to provide the model with information on what type of input data is to follow.

I am a bit confused by the problem formulation and implementation:
- Multi-task learning works best if there are clear dependencies and shared characteristics between the task - I am not sure if the tasks used here (POS tagging, image captioning, VQA, video activity recognition) share sufficient characteristics
- Successful multi-task learning often leads to shared representations developing across the different tasks. Is there any indication of this happening in the proposed work? Would it help to make the modality specific encoders (BPE and ResNet) trainable, so that the transformer model has an easier time learning multi- and cross-modal representations? Or is this entirely the CNP's role?
- It is probably possible to use HogWild as a way to train a multi-modal and multi-task system in the described manner, but will this lead to shared representations developing? Is there any way to include regularization or other tricks to encourage the formation of shared cross-modal representations?

Here are a few ideas:
- Would it be possible to show how well the model works when trained and optimized (including hyper parameter optimization) on a single task only? it would then be possible to show that the multi-task model can outperform single-modality single-task models
- What about tasks like speech input? There is other, published work on multi-task training for CTC models w.g. by Ramon Sanabria ("Hierarchical Multi Task Learning With CTC", SLT 2018), in addition to the reference provided
- Would it be possible to show how the model learns to develop cross-modal representations during training? this would be a convincing visualization and rationalization of the proposed approach


**Experience Assessment:**

I have published one or two papers in this area.

**Review Assessment: Checking Correctness Of Derivations And Theory:**

I assessed the sensibility of the derivations and theory.

**Review Assessment: Checking Correctness Of Experiments:**

I assessed the sensibility of the experiments.

**Review Assessment: Thoroughness In Paper Reading:**

I read the paper at least twice and used my best judgement in assessing the paper.

---

> ### Author Response · Authors · 2019-11-13
> **Author Response to Reviewer #1**
>
> >>>> Multi-task learning works best if there are clear dependencies and shared …
>
> We agree that the benefits of tasks are more relevant when there are clear dependencies and shared characteristics between the tasks. However, the primary objective of the paper is to verify the applicability of the proposed generic CNP architecture to support any task with spatio-temporal multi-modal data. To verify this, we individually evaluate the architecture on all possible spatio-temporal archetypes.
> 1.  Single input with (t > 1 & s = 1): POS tagging task which is by default supported by transformers as it contains only temporal modality.
> 2. Single input with (t = 1 & s > 1): Image captioning task demonstrates the capability of the model to attend on spatial data.
> 3. Multiple inputs, each with either (t > 1 & s = 1) or (t = 1 & s > 1): The third task, Visual Question Answering demonstrates the capability of the model to attend on multiple inputs with spatial or temporal modalities.
> 4. Single input with (t > 1 & s > 1): Finally, HMDB verifies the ability of the model to attend on inputs containing spatio-temporal information.
> Each of the tasks intends to verify the applicability of the architecture on a specific spatio-temporal input combination.
> Next, we demonstrate the model is able to learn all of the four above mentioned tasks in a multi-task setting without any reduction in performance. However, since the tasks are not completely related, we do not observe a jump in performance in a multi-task setting
>
>
> >>>> Successful multi-task learning often leads to shared representations ...
>
> Yes, we agree, successful multi-task learning should help in developing shared representations across multiple tasks. The ablation study and the results in zero-shot learning indicate this. The results of the ablation of spatial cache for image captioning task demonstrate that in a single task setting the performance drops to zero. However, when trained in a multi-task setting ablation of spatial cache still retains some performance for image captioning. As other tasks utilize the temporal cache for prediction, in the multi-task setting the captioning task learns to utilize the spatial average of the image stored in the temporal cache for prediction and hence retains some performance after ablation of the spatial cache. This demonstrates that the mechanisms learned for prediction during training are indeed shared during the multi-task setting.
>
> We also demonstrate zero-shot learning capabilities with the multi-task learning model. The multi-task model trained on image captioning, image QA and video action recognition can perform video captioning and video QA without any explicit training on these tasks. This demonstrates that in the multi-task scenario the model can leverage learning from multiple tasks to perform tasks that were never trained. It used the learned representations from image captioning and can apply the same on videos, thereby demonstrating the effectiveness of the shared representations learned by the CNP.
>
> Certainly, keeping the modality-specific encoders trainable should further help in learning better shared representations for tasks that use the same set of peripherals in their input. The role of the CNP is to work directly with spatio-temporal tensor representations of the inputs generated by the peripherals. In our paper, we did keep the language peripheral (BPE layers) trainable and freeze the vision peripheral (ResNet layers). However, it would be ideal to keep all the peripherals trainable.
>
>
> >>>> It is probably possible to use HogWild as a way to train a multi-modal ...
>
> Our study excludes the much important and investigated the area of scheduling mechanisms for multiple tasks and their inputs. We use Hogwild as a training tool for faster asynchronous computations and adding a regularization or load sharing strategy could help in achieving desired sharing across tasks and modalities.
>
>
> >>>> a few ideas: - Would it be possible to show how …
>
> We thank the reviewer for really interesting set of ideas. As suggested, we will try the hyperparameter optimization results for individual tasks and can add them in the final version of the paper. In the future, it would be interesting to add support for more modalities like speech and add more tasks to the setup. We have also open-sourced the code for our architecture along with the first results to allow the community to re-use and assist in its further development.

---

### Official Review · AnonReviewer2 · 2019-10-22
**Official Blind Review #2**

**Rating:** 6

**Review:**

This paper proposes a unified architecture in the context of multi-task learning where they demonstrate that training four tasks (with a variety of modalities like image, text, and videos) together results in about three times compressed model, while maintaining the performance similar to their respective individually trained models. The major components of this archtiecutre are (1) peripheral networks: used to encode the domain specific input into feature representations. (2) central neural processor: a fully attention based encoder-decoder model similar to the Transformer networks which encodes the spatio-temporal information. Further, this paper suggests that their unified architecture enables to perform decent on unseen tasks during its training. In this paper they test such scenarios on video captioning and video question answering.

Overall, the paper is clear to read and thorough in its experiments, with the caveat that its missing many multi-task paper references and the ideas are not much novel. However, I would say the setup is well engineered.

Arguments:

1) There are many works in multi-task learning after Luong et al., 2016, please refer them in the related work section (this section is very short!) and discuss on the differences of your model w.r.t. previous work (this is completely missing in the paper). I am pointing to some references below.

2) Statistical significance tests are missing to show that MULT-3 or MULT-4 are able to “maintain” the performance w.r.t. IND.


[1] Latent Multi-task Architecture Learning, Ruder et al., 2019
[2] A Joint Many-Task Model: Growing a Neural Network for Multiple NLP Tasks, Hashimoto et al., 2017
[3] Multi-Task Video Captioning with Video and Entailment Generation, Pasunuru & Bansal, 2017


**Experience Assessment:**

I have published one or two papers in this area.

**Review Assessment: Checking Correctness Of Derivations And Theory:**

I assessed the sensibility of the derivations and theory.

**Review Assessment: Checking Correctness Of Experiments:**

I carefully checked the experiments.

**Review Assessment: Thoroughness In Paper Reading:**

I read the paper thoroughly.

---

> ### Author Response · Authors · 2019-11-14
> **Author Response to Review #2**
>
> We thank the reviewer for encouraging feedback and comments. As suggested, we will add a  discussion on multi-tasking literature in the final version of the paper.  We are in the process of hyper-parameter optimization as pointed by Reviewer#1 and will perform the detailed statistical significance tests for comparison of MULT-3 and MULT-4 with IND. We will add these results as well to the final version of the paper.

---

### Official Review · AnonReviewer3 · 2019-10-25
**Official Blind Review #3**

**Rating:** 3

**Review:**

The authors propose an extended and unifying learning architecture – OmniNet- based on transformer, which tackles tasks with various modalities such as images, text and videos. This is attained with a spatio-temporal cache mechanism, that can both capture and store temporal information and spatial information. In addition, this proposed framework supports asynchronous multi-task learning with pre-trained neural networks on different modalities. OmiNet’s generalization capability is illustrated and demonstrated with experiments. The proposed model has multiple peripheral networks each majoring on one unique modality of data. These peripheral networks project input data into the same shared format/space that can be uniformly processed in a central neural processor working like a CPU. The central neural processor uses self-attention and RNN for temporal encoding and spatial encoding. The output of the encoding components are stored in temporal and spatial caches respectively. The two caches are then used as input to the spatial temporal decoder for different tasks.


Overall, this paper is well-written, and technically sounds, with comprehensive experimental results. However, I still have two concerns below that prevent me from giving a direct acceptance.


1.	However, considering the proposed model attempts to solve the multi-task learning problem, there seems no multi-task learning methods compared as baselines, making it hard to justify the performance.

2.	Furthermore, in Table 1, the Omin-Net results are not as good as SOTA, without clear explanation. The parameter settings for SOTA are also missing.


**Experience Assessment:**

I have published one or two papers in this area.

**Review Assessment: Checking Correctness Of Derivations And Theory:**

I carefully checked the derivations and theory.

**Review Assessment: Checking Correctness Of Experiments:**

I carefully checked the experiments.

**Review Assessment: Thoroughness In Paper Reading:**

I read the paper thoroughly.

---

> ### Author Response · Authors · 2019-11-07
> **Author Response to Review #3**
>
> We thank the reviewer for their time and detailed feedback. We have detailed responses to your concerns individually below:
>
> Answer to 1:-
> --------------------
> Most of the existing baselines in multi-task learning consider a set of related tasks. Therefore, they are designed to have a specific model that is highly biased towards the chosen tasks or their input domains. For example, work done in [1] demonstrates an architecture capable of learning a shared representation across audio and video modalities. Similarly, in recent work [4], a unified Transformer based approach has been proposed for all text-based tasks. To the best of our knowledge, there is no single architecture in the multi-task learning literature that unifies multi-modal learning across multiple tasks.  Our work, on the other hand, proposes a generic architecture for the unification of multi-modal learning by enabling the architecture (using CNP) to support any number of spatio-temporal input modalities.
>
> Therefore, in the absence of any baseline (for multi-tasking on multi-modal tasks), we are unable to compare our model directly with any existing work. We instead compare our results with the individual state-of-the-art across these tasks. We present our results as the first baseline for a unified multi-modal multi-task model. Interestingly, our multi-tasking framework comes with another benefit of zero-shot learning of untrained tasks such as video captioning and video question answering.
>
>
> Answer to 2:
> ------------------
> In our work, we have a generic architecture that can be used across different tasks of various modalities. On the other hand, most of the SOTA architectures follow a very specific approach to solve accurately the task in hand:
>
> 1. The SOTA paper for VQA & Captioning [2] uses a combined bottom-up and top-down visual attention mechanism. In fact, they also leverage object detection models such as faster R-CNN to better attend on specific regions of an image.
> 2. The SOTA paper for HMDB [3] uses a novel two-stream convolutional architecture where the spatial stream performs action recognition from still video frames, whilst the temporal stream is trained to recognize action from motion in the form of dense optical flow.
> These architectures are specifically designed and tuned to accurately learn the representations for the task in hand. However, due to the lack of task-specific design and optimization in our generic architecture, we do not obtain SOTA scores across all the tasks. The focus of this work is on methodological contribution towards the unification of learning across multiple domains instead of beating the SOTA. We set the default hyperparameters as that in the Transformer paper [5], due to resource constraints. Thus, although we obtain reasonable scores in multi-task or single task settings, our architecture does not achieve SOTA in each of the tasks. With task-specific peripherals and hyperparameter search, we believe the results would be comparable to that of SOTA.
> The results of SOTA are directly referenced from corresponding citations and thus, have the parameter settings as detailed in them.
>
>
> REFERENCES:
> [1] Multimodal deep learning, Ngiam et al., 2011
> [2] Bottom-Up and Top-Down Attention for Image Captioning and Visual Question Answering, Anderson et al., 2017
> [3] Two-Stream Convolutional Networks for Action Recognition in Videos, K Simonyan,  ‎2014
> [4] Exploring the Limits of Transfer Learning with a Unified Text-to-Text Transformer. Transfer learning, C Raffel et al, 2019
> [5] Attention is all you need, A Vaswani et al, ‎2017

---

> > ### Comment · AnonReviewer3 · 2019-11-15
> > **Responses to authors**
> >
> > Response to Author Response #1:
> >
> > Thanks for the authors’ responses. I agree that most existing methods in multi-task learning consider a set of related tasks. However, if OmniNet works for multi-task learning on multi-modal tasks, it should naturally tackle the traditional single-modal multi-task learning (single module) problems, e.g., in [1]? Moreover, there are papers talking about multi-task learning on multi-modal tasks, such as [2, 3]. Therefore, if I did not miss something, these baselines should all be included and compared with OmniNet for completeness of the proposed work.
> >
> > [1] Asynchronous Multi-task Learning. https://arxiv.org/pdf/1609.09563.pdf
> > [2] MultiNet: Multi-Modal Multi-Task Learning for Autonomous Driving. https://arxiv.org/pdf/1709.05581.pdf
> > [3] Multi-task Learning for Multi-modal Emotion Recognition and Sentiment Analysis. https://arxiv.org/pdf/1905.05812.pdf
> >
> > Response to Author Response #2:
> >
> > The proposed method is an interesting approach towards the unification of learning across multiple domains. However, without seeing its full potential (after well trained and with optimized hyper-parameters), it is difficult to judge if OmniNet will outperform single task models.
> >
> > Please let me know if I missed anything here.

---

> > > ### Author Response · Authors · 2019-11-15
> > > **Response to Reviewer #3**
> > >
> > > Thank you for addressing our responses!
> > >
> > > We agree that the model can be compared with existing multi-task settings. However, our choice of tasks was targeted to verify the applicability of the proposed generic CNP architecture to support a single task with spatio-temporal multi-modal data. Therefore, we individually evaluate the architecture on all possible spatio-temporal archetypes.
> > > 1.  Single input with (t > 1 & s = 1): POS tagging task which is by default supported by transformers as it contains only temporal modality.
> > > 2. Single input with (t = 1 & s > 1): Image captioning task demonstrates the capability of the model to attend on spatial data.
> > > 3. Multiple inputs, each with either (t > 1 & s = 1) or (t = 1 & s > 1): The third task, Visual Question Answering demonstrates the capability of the model to attend on multiple inputs with spatial or temporal modalities.
> > > 4. Single input with (t > 1 & s > 1): Finally, HMDB verifies the ability of the model to attend on inputs containing spatio-temporal information.
> > > Each of the tasks intends to verify the applicability of the architecture on a specific spatio-temporal input combination.
> > >
> > > Next, we demonstrate the multi-task capability for model to learn all of the four above mentioned tasks in a multi-task setting without any reduction in performance.
> > >
> > > To the best of our knowledge, our work is the first attempt to enable generic multi-modal learning, along with multi-tasking and zero-short learning capabilities. Further, we achieve comparable results with the default set of hyperparameters. We agree that a deeper training (under progress) could lead to even better results.
> > > We hence summarize the major contributions and novelty of our paper as:-
> > > 1. A unified architecture capable of learning tasks with any combination of input domains.
> > > 2. A model capable of multi-task learning without any reduction in performance
> > > 3. Zero-shot learning capability on untrained tasks in a multi-task setting.
> > >
> > > We hope this clarifies the reasoning behind the chosen setup of tasks and the importance of the presented architecture and first such results in the direction of generalizing multi-modal learning with multi-tasking and zero-short learning capabilities.

---

### Author Response · Authors · 2019-11-14
**Discussion**

We thank the reviewers for their detailed reviews and many valuable comments. We would also appreciate your attention to the responses. Do let us know if any of your concerns that are still unaddressed.

---

### Decision · Program_Chairs · 2019-12-19

**Decision:**

Reject

**Comment:**

This paper presents OmniNet, an architecture based on the popular transformer for learning on data from multiple modalities and predicting on multiple tasks.  The reviewers found the paper well written, technically sound and empirically thorough.  However, overall the scores fell below the bar for acceptance and none of the reviewers felt strongly enough to 'champion' the paper for acceptance.  The primary concern cited by the reviewers was a lack of strong baselines, i.e. comparison to other methods for multi-task learning.  Unfortunately, as such the recommendation is to reject.  However, adding a thorough comparison to existing literature empirically and in the related work would make this a much stronger submission to a future conference.